# Locomotor Strategy to Perform 6-Minute Walk Test in People with Multiple Sclerosis: A Prospective Observational Study

**DOI:** 10.3390/s23073407

**Published:** 2023-03-24

**Authors:** Nawale Hadouiri, Elisabeth Monnet, Arnaud Gouelle, Yoshimasa Sagawa, Pierre Decavel

**Affiliations:** 1Laboratory of Clinical Functional Exploration of Movement, University Hospital of Besançon, 25000 Besançon, France; 2Clinical Investigation Center, INSERM 1431, University Hospital of Besançon, 25000 Besançon, France; 3Department of Physical Medicine and Rehabilitation, Dijon-Bourgogne University Hospital, 21000 Dijon, France; 4EA4266 Agents Pathogènes et Inflammation, University of Bourgogne-Franche-Comte, 25000 Besançon, France; 5Laboratory Performance, Santé, Métrologie, Société (PSMS), UFR STAPS, 51000 Reims, France; 6Integrative and Clinical Neurosciences EA481, Bourgogne Franche-Comte University, 25000 Besançon, France; 7Rehabilitation Department, HFR, 1700 Fribourg, Switzerland

**Keywords:** 6 min walk test, walking, Multiple Sclerosis, physical adaptation, spatio-temporal parameters

## Abstract

Two-thirds of people with Multiple Sclerosis (PwMS) have walking disabilities. Considering the literature, prolonged tests, such as the 6 min walk test, better reflect their everyday life walking capacities and endurance. However, in most studies, only the distance traveled during the 6MWT was measured. This study aims to analyze spatio-temporal (ST) walking patterns of PwMS and healthy people in the 6MWT. Participants performed a 6MWT with measures of five ST variables during three 1 min intervals (initial: 0′–1′, middle: 2′30″–3′30″, end: 5′–6′) of the 6MWT, using the GAITRite system. Forty-five PwMS and 24 healthy people were included. We observed in PwMS significant changes between initial and final intervals for all ST parameters, whereas healthy people had a rebound pattern but the changes between intervals were rather negligible. Moreover, ST variables’ changes were superior to the standard measurement error only for PwMS between initial and final intervals for all ST parameters. This result suggests that the modification in PwMS’ walking pattern is effectively due to their walking ability and not to a measurement, and suggests that PwMS could not manage their walking efficiently compared to healthy people, who could maintain their rhythm throughout the 6MWT. Further studies are needed to detect these patterns changes in the early evolution of the disease, identify clinical determinants involved in PwMS’ walking pattern, and investigate whether interventions can positively impact this pattern.

## 1. Introduction

Multiple Sclerosis (MS) is a chronic inflammatory autoimmune disease of the central nervous system [1]. MS represents the main neurological cause of young adult functional disability [2]. The symptoms of MS are variable, but most are sensory, cognitive, or motor impairments. This has a negative impact on the activity and participation of People with MS (PwMS) with disabilities affecting daily life activities, e.g., walking, gripping, speaking, having good bladder and bowel control or a satisfying sexual life, or even visual, thinking, memory, and mood disabilities, following the nomenclature of International Classification of Functioning Disability and Health [3,4,5]. Among them, the walking disability is the most limiting factor for daily life activities in most PwMS [3,6]. This is also the reason why a walking disability is a significant criterion to assess the MS progression in the Expanded Disability Severity Scale (EDSS) [7]. The EDSS has been considered as the gold standard for diagnosing the clinical and functional severity of MS [7]. However, recent studies have highlighted its limitations and suggested that this scale should be supplemented with objective standardized measurements [8,9].

Different walking evaluations have been proposed and used in research and clinical practice to explore walking disorders in PwMS. Collectively, these tests can be categorized as short tests (e.g., the timed 25-foot walk [10] and 10-meter test [11]) or prolonged tests (e.g., the 6-minute walk test; 6MWT [12]). They are usually applied to evaluate maximal walking speed [13] and walking endurance [12], respectively. In complement to the tests mentioned above, spatio-temporal (ST) walking variables (e.g., velocity, cadence, stride length, stride width, base of support, etc.) have been used in many studies to assess in more detail walking disorders of PwMS in short tests [14,15,16,17,18]. Based on these evaluations, the main alterations found were that PwMS walked with a slower speed, decreased cadence and stride length, while having a wider base of support and a prolonged double support time compared to healthy people [14,15,16,17,18].

Although there is no clear consensus on which test should be used to assess walking in PwMS, the 6-minute walk test (6MWT) was recently recommended to evaluate gait in PwMS because it highlights motor fatigue resulting from extended task execution, thus, effectively assessing the physical efforts and level of autonomy of PwMS. For these reasons, prolonged tests such as the 6MWT are a better indicator of the ability of PwMS to perform the activities of daily living [19,20]. Initially, the 6MWT was a commonly used clinical assessment of exercise capacity in patients with cardiopulmonary diseases [21,22], and was then validated in MS. Additionally, since 2018 it is recommended in a clinical guideline to assess MS [11,12]. The main limitation of the 6MWT is that only the overall distance traveled is measured and explored in most articles on MS [11]. Therefore, it is difficult to understand the locomotor strategies PwMS use to manage their walking throughout this test. In this sense, and to bring new insights, analysis of spatio-temporal (ST) parameters during specific intervals of the 6MWT has been used in recent studies [23,24,25,26,27]. This approach seems applicable and comprehensive in clinical practice and may detect a decrease in motor possibilities to manage prolonged walking.

In this context, we published a first study based on evaluating the test–retest reliability of the ST parameter measures in some key intervals of the 6MWT in PwMS, and found them to be reliable (e.g., intraclass correlation coefficient (ICC) range for PwMS: 0.858–0.919) [25]. Moreover, we observed that PwMS had a different ST walking pattern compared to healthy people (e.g., PwMS had a “constant decline” pattern, i.e., a monotonous decrease in velocity was observed during the 6MWT, whereas a “rebound” pattern was found in the healthy group). These results corroborate and complete previous studies in which only walking velocity was analyzed in PwMS [12,28]. Because the first reliability step was checked and to complementarily confirm the clinical pertinence of our previous results, this study aimed to focus on the analysis of the difference between the ST walking pattern of PwMS and healthy people during specific intervals in the 6MWT. We expect that the assessment of other ST parameters measured during specific key intervals of the 6MWT, which could be easy to use in clinical routine practice, will allow us to understand the effects of loss compensation during prolonged walking conditions and will permit us to determine walking dysfunction earlier than classic analysis in degenerative disease. Our hypotheses were that (i) PwMS had a significant deterioration pattern in terms of ST parameters during the 6MWT intervals compared with comparable healthy people and (ii) in PwMS, the gait pattern could depend on MS phenotype and disease severity.

## 2. Methods

### 2.1. Study Design

This study was an ancillary study from the FAMPISEP project (NCT02849782). This project was a single-center phase 4 trial. It was conducted between April 2014 and March 2019 at the University Hospital of Besançon (France) to investigate the effects of fampridine (4-aminopyridine) in PwMS with walking disorders. However, in this ancillary study, Fampridine action was not evaluated.

The protocol was governed by French legislation concerning interventional biomedical research and had been submitted to the National Ethics Committee (#13/405). The study was approved by the French Health Products Safety Agency (#2013-A002305-56). After providing the participant with information about the procedure and ensuring this information was adequately understood, written informed consent was obtained from all participants of this study.

### 2.2. Participants

All participants were examined by the same practitioner to check the inclusion and exclusion criteria and to document the participant’s characteristics. All participants did not suffer from any significant chronic cardiorespiratory disease.

For PwMS, the inclusion criteria were:(i)an MS diagnosis according to the modified McDonald criteria [29];(ii)an EDSS score of 4.0–6.5. We included PwMS in this range because it is between the values of 4.0 and 6.5 in which gait worsening is rated [7]. According to the literature, PwMS with an EDSS of 4 had moderate MS. PwMS with an EDSS between 4.5 and 6.5 had severe MS [12];(iii)the capacity to walk for at least 6 min (according to the data from the medical examination, i.e., EDSS).

The exclusion criteria were:(i)worsening MS symptoms during the previous 60 days;(ii)immunotherapy change in the previous 60 days;(iii)the presence of other neurological disorders.

A group of healthy volunteers (healthy group) was recruited from the general community to make a comparable group with PwMS in terms of distribution of sex, age, body mass, body height, and body mass index (BMI); they presented no neuro-orthopedic problems or other antecedents that could compromise their walking capacities.

For feasibility and recruitment of healthy people, a 2:1 ratio (i.e., 2 PwMS for 1 healthy person) was performed.

## 3. Evaluations

### 3.1. Clinical Evaluation

For PwMS, a neurological examination was performed to determine their disability level through the Expanded Disability Status Scale (EDSS), disease duration, and MS phenotype (relapsing-remitting [RR], secondary progressive [SP], or primary progressive [PP]).

### 3.2. Walking Evaluation

Participants performed two 6MWTs (test and retest) one week apart in our laboratory. Both tests were performed following the protocol of the American Thoracic Society in both groups [21], with the exception that the participants performed the test on a 24 m circuit rather than the recommended 32 m circuit due to local architectural constraints. During the 6MWT, foot contacts were recorded by a 6.10 m by 0.61 m GAITRite electronic walkway (CIR Systems, Franklin, NJ, USA) (Figure 1A). At the same time, the software PKMAS (ProtoKinetics, Havertown, PA, USA) was used to process and export the following ST variables for each walkway passage: velocity (m/s), cadence (steps/min), stride length (m), stride width (m), and double support time (% of the gait cycle) (Figure 1B). These measurements were recorded during three distinct intervals of 1 min of the 6MWT but were not continuous during the 1 min intervals and only occurred when the participants crossed the GAITRite [30]. Walking assessments in these specific intervals respect the number of steps required to study the ST variables as mentioned above [30]. The studied intervals were: “*initial*” (from the start (0 s) to the end of the first minute (60 s); “*middle*” (from 2.5 min (150 s) to 3.5 min (210 s)); and “*end*” (from 5 min (300 s) to the end of 6 min (360 s)) (Figure 1B). These 1 min intervals were chosen to offer a more discriminant insight into the analysis of strategies adopted during the 6MWT while keeping appropriate levels of applicability and comprehensiveness for clinical practice [24,25]. To evaluate effort, a 15-grade Rating Perception Exertion Borg scale was performed verbally, without interrupting the 6MWT at the end of each interval [31], and the total distance traveled at the end of the 6MWT was measured.

### 3.3. Statistical Analysis

Due to the aim of our previous study, the power was verified for the available sample of 45 PwMS concerning the evaluation criterion reliability of the ST pattern during the 6MWT. Assuming a minimal ICC of 0.5 against a desired of 0.8 based on α = 0.05, n (number of observations) = 2, the power was 99.3% [32].

For data analysis, we used the average of both test and retest values (assessments one week apart). Since velocity differed between groups and the literature shows that velocity is strongly related to ST parameters, ST parameter values were expressed in changes (%) in relation to the values observed in the initial interval, for example:



(1)
$${\mathrm{Change}\;\mathrm{of}\;\mathrm{Velocity}}_{\mathrm{middle}/\mathrm{initial}}\;(\%)\;=\;100\times \frac{\left(Velocity\;(middle)\;-\;velocity\;(initial\right)}{velocity\;(initial)}$$



The Standard Error of Measurement was calculated as follows:(2)$$\mathrm{SEM}\;(\%)=100\times \sqrt{1-ICC}\times {(SD}_{Velocity\;\left(middle\right)\;-\;velocity\;\left(initial\right)}/{SD}_{Velocity\;\left(initial\right)})$$

Using the intraclass correlation coefficients (*ICC*) from our precedent test–retest reliability study) [25,33] for each ST parameter between middle/initial and final/initial intervals in each group to evaluate if the changes observed were greater than the measurement error.

For statistical analysis, comparisons within groups (i.e., PwMS middle/initial vs. PwMS final/initial) and between groups (i.e., PwMS middle/initial vs. Healthy people middle/initial) were conducted using Student’s t-tests (paired or independent) or their non-parametric equivalents when necessary (i.e., Wilcoxon matched pairs test or Mann–Whitney U test). A Bonferroni correction was applied to counteract the multiple comparisons problems with n = number of comparisons = 2, the α value for each comparison equal to 0.025.

## 4. Results

Forty-five PwMS and twenty-four healthy people were included in the study (Table 1). The ST variables recorded during each interval of the 6MWT, perceived exertion measured by Borg scale, and overall distance covered are detailed in Table 2 for both groups. PwMS covered a significantly shorter distance (−45%) during the 6MWT compared to healthy people (mean (SD): PwMS 358 (99) m vs. Healthy 655 (75) m; *p* < 0.0001).

In PwMS, a monotonous decrease was observed for velocity and cadence. The stride length decreased between the middle and final intervals. Stride width only increased between the initial and middle interval and double support showed a constant increase during the 6MWT.

In the healthy group, velocity, cadence, and stride length decreased between the initial and middle intervals and increased between the middle and final intervals. No major variation was shown for stride width between the intervals except a small decrease from the initial to the middle interval. The double support showed an increase between the initial and middle intervals and a decrease between the middle and final intervals.

Table 3 summarizes the changes and standard error of measurement between middle/initial and final/initial in each group (e.g., PwMS and healthy groups) and between groups.

Changes (%) and SEM (%) were observed for the middle and final intervals in relation to the initial interval. In PwMS, statistically significant deteriorations were observed for all ST variables. In relation to the initial interval and for all ST variables, changes varied between 2% and 9%. Changes were superior to the SEM only in the PwMS group between the initial and final intervals and ranged between 2.6% and 8.4% for this group and intervals (Table 3 and Figure 2).

Regarding changes in the healthy group, small deteriorations of velocity, cadence, and double support time were observed between the initial and middle intervals. The healthy group preserved their stride length and stride width throughout the 6MWT with no statistically significant changes (Table 3). In relation to initial condition and for all ST variables, changes varied between 0% and 3% and were inferior to SEM in all interval comparisons (SEM values ranged from 2.5% to 7.1%) (Table 3 and Figure 2).

Regarding comparisons of changes between groups, statistically significant differences were present for all ST parameters (*p*-values < 0.01) except for stride length and double support time for a change in the middle/initial interval.

To better visualize the ST parameters evolution over these 6MWT intervals in both groups, a graphical representation was provided for the averaged ST variables (during each interval of the 6MWT) in Figure 3.

In sub-groups of PwMS, we could observe some trends of the descriptive course of the ST walking pattern in the intervals of the 6MWT according to the EDSS status (Table 4 and Figure 4) and according to MS phenotype (Table 5 and Figure 5).

We observed a more pronounced monotonous decrease in velocity between the middle and end intervals in patients with severe EDSS compared to moderate EDSS. Velocity, cadence, and stride length were lower in each interval in those with severe EDSS compared to the moderate EDSS group. In contrast, stride width and double support time were higher in the severe EDSS group than in the moderate EDSS group.

Considering the MS phenotype, velocity, cadence, and stride length in the SP group showed a more marked decrease than the other two groups, which showed a monotonous decrease. Velocity, cadence, and stride length were lower in each interval in the PP group than in the other two groups, while stride width and double support time were higher in the PP group than in the other two groups. Velocity, cadence, and stride length were lower in each interval in the SP group than in the RR group, while stride width and double support time were higher in the SP group than in the RR group.

## 5. Discussion

In our present study, we aimed to explore how PwMS manage a prolonged effort by examining the variations of spatio-temporal variables during specific intervals on one of the most widely used prolonged walking tests, namely the 6MWT. The distance traveled by the PwMS group was close to values previously reported in PwMS with a similar EDSS, suggesting the representativeness of our evaluation [12,16,20,34].

Here, despite the fact that the 6MWT velocity or distance traveled in PwMS was within the normal limits values of healthy people [35], we observed that they employed a different strategy to perform the 6MWT compared to their healthy peers according to their ST walking pattern. PwMS seemed to have a significantly altered walking pattern (i.e., a constant decline pattern) compared to healthy people (who display a rebound pattern).

On the one hand, in PwMS, we observed significant changes between initial and final intervals for all ST parameters. The ST variables’ changes were superior to SEM only for PwMS between initial and final intervals for all ST parameters. This result suggests that the modification in PwMS’ walking pattern is effectively due to their walk disability, not to a measurement error and indicates that PwMS could not manage their walking efficiently. The observed decrease in velocity could be linked to its two components: cadence and stride length. Usually, in the beginning of pathological situations, a shortening of stride length is first observed. Then, due to disease progression, a decrease in velocity appears caused by the loss of the ability to increase the cadence [36,37]. We can hypothesize that moderate to severe PwMS (i.e., with an EDSS between 4.0 and 6.5), contrary to healthy people, had lost their ability to compensate the decrease in step length by increasing cadence, for example, due to fatigue or central nerve conduction disturbance [38]. The 6MWT could be considered in PwMS as a maximal effort test according to the literature [25,39]. In our present study, according to their median EDSS, PwMS have almost traveled in 6 min the distance they can best do without time constraints (e.g., 358 m during the 6MWT compared to 300–500 m with no time restriction in PwMS with an EDSS score between 4 and 5). PwMS presented in our study (i) a constant decline pattern with significant changes between initial and final intervals of the 6MWT, which were superior to SEM and (ii) higher scores on the Borg scale, so a perceived exertion more negative than healthy people. The Borg scale was designed to assess the strenuousness of a physical activity in resistance, which is not the case for walking. However, there is an increased exertion in MS patients when compared to healthy subjects during a submaximal task such as the 6-minute test. Nevertheless, we had shown that the reliability of the Borg scale was not good in MS subjects in our previous article [25]. The ST pattern of the 6MWT and the least good tolerance to physical exertion during this test could also be due to secondary cardiorespiratory deconditioning from the sedentary lifestyle adopted due to the symptoms of MS [40,41].

In PwMS, stride width and double support time increased during the 6MWT. This could be correlated with the decrease in walking velocity, cadence, and stride length, perhaps to avoid falling and might reflect a loss of walking stability [42,43]. Our results corroborate previous studies, which found a decrease in velocity and an increase in double support time [26,44]. These results underlined the importance of integrating variables that reflect walking stability to better understand the overall ST pattern during the 6MWT in PwMS.

To investigate the PwMS ST pattern, the impact of some MS characteristics has been studied here, such as disease severity (EDSS) or MS phenotype. Unfortunately, the present PwMS sample size (and PwMS subgroup sample size) precluded multivariate analysis of the relationship between ST walking patterns and MS characteristics, such as EDSS, disease duration, MS phenotype, fatigue, or further statistical analysis inside each subgroup of PwMS. However, we observed certain trends in these subgroups (Appendix A). According to these results, we could hypothesize that EDSS and MS phenotype could impact their ST pattern. Concerning EDSS, two studies in the literature analyzed velocity or distance traveled during the 6MWT (with a different temporal segmentation) in PwMS with mild (EDSS between 0 and 2.5), moderate (EDSS between 3.0 and 4.0), and severe (EDSS between 4.5 and 6.5) MS. In these two studies, as in our study, the greater the EDSS, the more PwMS seemed to display a *monotonous decrease* pattern in velocity or distance traveled during the 6MWT [12,20]. However, the current sample size did not allow us to study the relationship between the ST walking pattern and MS characteristics. Further studies are needed to determine the contribution of each MS determinant to the ST altered pattern (i.e., the impact of MS phenotype in the *rebound* pattern loss).

On the other hand, in healthy people, we observed a rebound pattern as already described in the literature [12,25,28], but we highlighted here that the ST variables’ changes between 6MWT intervals were negligible, i.e., healthy people could maintain their rhythm throughout the 6MWT. Healthy people seemed to conserve abilities to maintain velocity by modulating their two components of velocity: cadence and stride length (Table 2). The Borg exertion perception values were also lower in the healthy people compared to PwMS and they were stable throughout the 6MWT interval. Indeed, the 6MWT has been described as a submaximal effort test in healthy people [21] which could explain their capacity to maintain their rhythm from a cardiorespiratory perspective with minor changes during the 6MWT and no significant changes of ST parameters during the 6MWT, and with SEM superior to these changes.

These above elements partly explain the ST pattern differences observed between PwMS and healthy people, and that some MS characteristics could impact this ST pattern.

This ST pattern analysis presents several perspectives for clinical practice and research.

Firstly, it could be easy to use the walking assessment in PwMS in clinical routine practice. These elements may soon become more important and readily available with the expanding use of portable gait analysis devices that record spatio-temporal parameters [45].

Secondly, studying variations in ST parameters across and between these intervals, before and after interventions, could provide new data to assess the efficacy of physical or medical therapy (i.e., such as fampridine treatment [46]) in terms of physical exertion management. Accordingly, this would allow for the adaptation of interventional targets in clinical practice. In some studies concerning pulmonary diseases (e.g., pulmonary arterial hypertension) [47], the ventilatory threshold has been used to set the optimal exercise load. In our study, the ST walking pattern in PwMS could be utilized identically to set up walking exercises, for example, by stimulating PwMS to move from a constant decline pattern to a rebound pattern to improve their abilities to manage their effort.

### Limitations

Although this study provides new insights concerning prolonged walking tests, it presents some limitations. Measurement of ST variables during the 6MWT was not continuous over the circuit and was limited to the length of the instrumented walkway. Therefore, considering the total size of the circuit, only 25% of this length was explored by the GAITRite electronic walkway system. This technical issue could be circumvented by using inertial sensors or instrumented insoles that would enable ST variables to be estimated for all gait cycles over the course of the circuit with continuous acquisition [26,27,48] or in a more simple way for clinical practice is to assess walking velocity evaluating the distances traveled in each 6MWT interval (i.e., distance/60 s). This seems an easy way to evaluate velocity patterns in a clinical condition.

In our precedent test–retest reliability study [25], reliability values of perceived exertion with Borg scale in PwMS were fair (e.g., kappa values between 0.21 and 0.40), which is a limitation to study perceived exertion, hence here the interest to have other objective measures to improve our understanding of PwMS’ walking management throughout the 6MWT.

Finally, the sample size was a limitation. Indeed, it was based on the previous study aim, i.e., it was a reliability study. Moreover, our sample size precluded any investigation of the relationship between ST walking pattern and MS characteristics. For future studies, it will be essential to consider the variation for a given criterion (e.g., mean of standard deviations for speed intervals) and a minimum expected change between intervals.

## 6. Conclusions

To conclude, the analysis of ST patterns throughout specific intervals on the 6MWT revealed that PwMS have altered walking patterns (i.e., a constant decline pattern) of velocity, cadence, and stride length compared to healthy persons who displayed a rebound pattern (without significant changes) during the 6MWT. These results reflect an altered walking management in PwMS despite the fact that the velocity or the 6MWT distance traveled were within the normal limits values of healthy people. In PwMS, stride width and double support time increased during the 6MWT. This could be correlated with the decrease in walking velocity, cadence, and stride length and reflect a loss of walking efficacy. These results underlined the importance of integrating variables that reflect walking stability to better understand the overall ST pattern during the 6MWT in PwMS. With increasing accessibility to wearable ST analysis systems, this type of analysis will be easy to use in clinical routine practice. Here, the potential association between retrieving patterns and the degree of impairment allows us to put this analysis into early detection of gait impairment. Further studies are needed to identify clinical determinants involved in this altered pattern and to investigate whether interventions can have a positive impact on this pattern (i.e., by stimulating PwMS to move from a constant decline pattern to a rebound or constant ST pattern) and not just focus on the improvement of the velocity or the 6MWT distance traveled to improve walking capacities of PwMS.

## Figures and Tables

**Figure 1 sensors-23-03407-f001:**
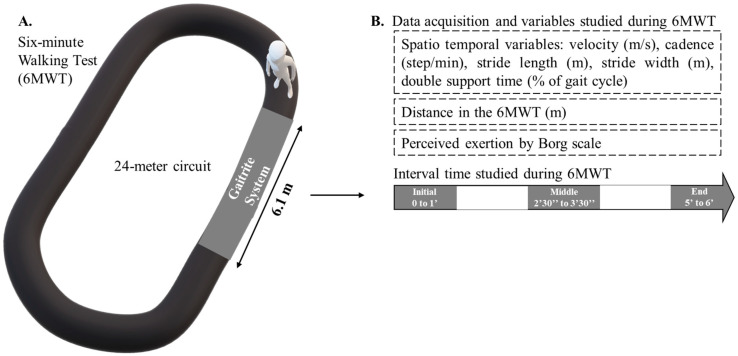
(**A**) gait evaluation during the six-minute walk test (6MWT); (**B**) variables studied during each interval of the 6MWT [25].

**Figure 2 sensors-23-03407-f002:**
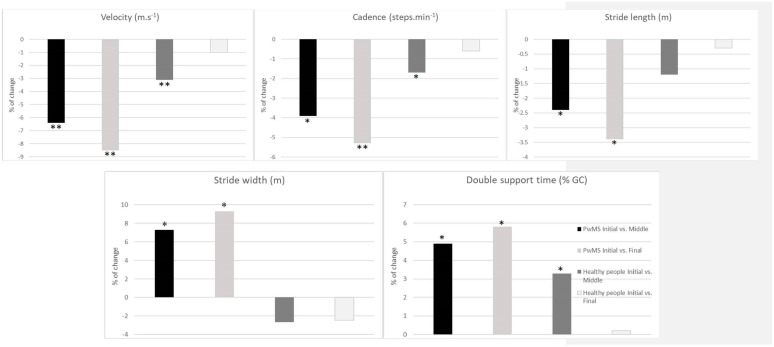
Changes between middle/initial and final/initial intervals in each group (e.g., PwMS and healthy groups). * < 0.01, ** < 0.001 significant difference in changes for the same group. GC, Gait Cycle.

**Figure 3 sensors-23-03407-f003:**
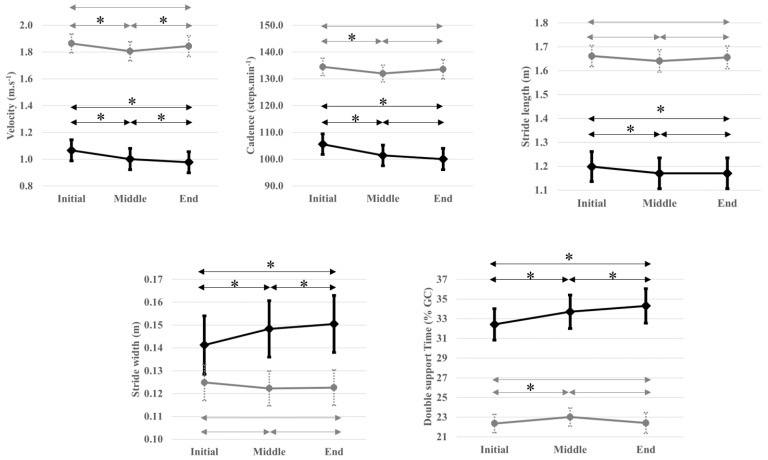
Spatio-temporal variables during each interval of the 6MWT for PwMS (black line) and healthy groups (grey line). GC, gait cycle. * Significant differences between intervals in each group (e.g., PwMS and healthy people) with *p* < 0.025.

**Figure 4 sensors-23-03407-f004:**
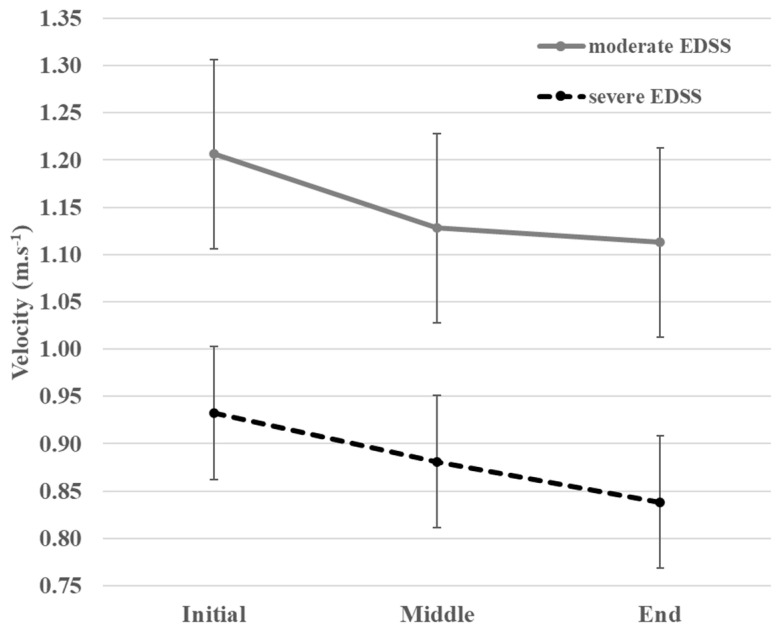
Variations of the velocity during the 6MWT for PwMS with moderate and severe EDSS. EDSS, Expanded Disability Status Scale.

**Figure 5 sensors-23-03407-f005:**
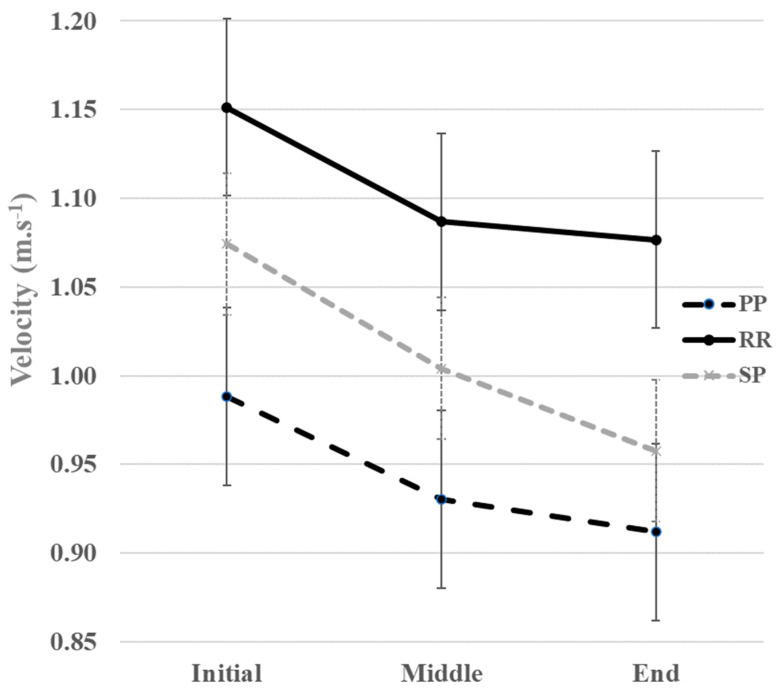
Variations of the velocity during the 6MWT for PwMS with PP, RR, and SP phenotypes. PP, Primary Progressive; RR, Relapsing–remitting; SP, Secondary Progressive.

**Table 1 sensors-23-03407-t001:** Characteristics of the study population.

	PwMS (n = 45)	Healthy (n = 24)	*p*
Gender (female/male; %)	26/19; 58/42	14/10; 58/42	0.812
Age (years) ^a^	51.4 (12.3)	51.3 (10.7)	0.22
Body mass (kg) ^a^	75.5 (18.3)	75.2 (13.7)	0.54
Body height (m) ^a^	1.7 (0.08)	1.7 (0.08)	0.78
BMI (kg·m^−2^) ^a^	26.2 (5.5)	26.1 (3.9)	0.27
Tobacco exposition (yes/no; %)	11/34; 32/68	5/19; 21/79	0.64
Alcohol exposition (yes/no; %)	4/41; 10/90	0/24; 0/100	0.09
EDSS (4–6.5) ^b^	4.5 [4;5]	NA	NA
Disease duration (years) ^a^	17.4 (8.9)	NA	NA
MS phenotype (n/%)		NA	NA
Relapsing-remitting	11/24	NA	NA
Secondary progressive	20/45	NA	NA
Primary progressive	14/31	NA	NA
EDSS per MS phenotype ^b^			
Relapsing-remitting	4 [4;4.125]	NA	NA
Secondary progressive	4.5 [4;5]	NA	NA
Primary progressive	5 [4;5.75]	NA	NA
Disease duration per MS phenotype ^b^			
Relapsing-remitting	13 [9;16]	NA	NA
Secondary progressive	19 [13.75;23]	NA	NA
Primary progressive	15 [8.75;20]	NA	NA

BMI, Body Mass Index; EDSS, Expanded Disability Status Scale; MS, Multiple Sclerosis; NA, Not Applicable; PwMS, People with Multiple Sclerosis; ^a^ mean (SD) and ^b^ median [1st quartile; 3rd quartile].

**Table 2 sensors-23-03407-t002:** Spatio-temporal walking parameters and perceived exertion during the three pre-defined one-minute intervals of the 6MWT for PwMS and healthy people.

	PwMS (n = 45)			Healthy (n = 24)		
	6MWT Intervals					
Variables	Initial	Middle	Final	Initial	Middle	Final
Velocity (m·s^−1^) ^a^	1.07 (0.27)	1.00 (0.27)	0.98 (0.27)	1.88 (0.22)	1.82 (0.22)	1.86 (0.25)
Cadence (step·min^−1^) ^a^	105.57 (13.17)	101.38 (13.23)	100.03 (13.51)	134.55 (8.90)	132.16 (9.00)	133.81 (10.82)
Stride length (m) ^a^	1.20 (0.21)	1.17 (0.22)	1.17 (0.22)	1.67 (0.13)	1.65 (0.14)	1.66 (0.14)
Stride width (m) ^a^	0.14 (0.04)	0.15 (0.04)	0.15 (0.04)	0.13 (0.02)	0.12 (0.02)	0.12 (0.02)
Double support time (% GC) ^a^	32.42 (5.49)	33.71 (5.83)	34.31 (5.99)	22.68 (2.67)	23.42 (2.74)	22.70 (3.24)
Borg scale (6; 20) ^b^	10 [9; 11]	12 [10; 13]	13 [12; 15]	9 [6; 10]	9 [8; 10]	9 [8; 10]
6MWT distance (m) ^a^	358 (99)			655 (75)		

GC, Gait Cycle; 6MWT, 6-Minute Walk Test; PwMS, People with Multiple Sclerosis;. ^a^ mean (SD) and ^b^ median [1st quartile; 3rd quartile].

**Table 3 sensors-23-03407-t003:** Changes and standard error of measurement between middle/initial and final/initial intervals in each group (e.g., PwMS and healthy groups) and between groups.

	PwMS (n = 45)		Healthy (n = 24)	
ST Variables ^a^	Middle/Initial	Final/Initial	Middle/Initial	Final/Initial
Velocity	−6.4 (5.5) **	−8.5 (7.7) **	−3.1 (3.3) ** ^†^	−1.0 (5.0) ^‡^
Cadence	−3.9 (3.4) *	−5.3 (4.6) **	−1.7 (1.8) * ^‡^	−0.6 (3.1) ^‡^
Stride length	−2.4 (3.3) *	−3.4 (4.5) *	−1.2 (2.1)	−0.3 (3.6) ^‡^
Stride width	7.3 (14.5) *	9.3 (18.8) *	−2.7 (6.8)	−2.5 (10) ^‡^
Double support	4.0 (3.9) *	5.8 (4.9) *	3.3 (3.6) * ^‡^	0.2 (6.8) ^†^
SEM (%)				
Velocity	9.3	7.1	6.6	7.1
Cadence	4.4	4.8	4.1	4.5
Stride length	2.5	2.6	2.9	3.5
Stride width	9.1	8.4	6.3	7.3
Double support	4.9	5.1	3.9	6.2

PwMS, People with Multiple Sclerosis; SEM, Standard Error of Measurement, ^a^ Change in % (SD), * < 0.01, ** < 0.001 significant difference in changes for the same group. ^†^ <0.01, ^‡^
*p* < 0.001 significant difference between groups for the same changes.

**Table 4 sensors-23-03407-t004:** Spatio-temporal variables during each interval of the 6MWT for PwMS with moderate and severe EDSS.

Moderate EDSS [4] (n = 22)				Severe EDSS [4.5–6.5] (n = 23)			
6MWT Interval							
ST Variables	Initial	Middle	End	ST Variables	Initial	Middle	End
Velocity (m·s^−1^)	1.21 (0.24)	1.13 (0.24)	1.11 (0.24)	Velocity (m·s^−1^)	0.93 (0.23)	0.88 (0.23)	0.84 (0.23)
Cadence (step·min^−1^)	111.32 (10.52)	107.93 (9.48)	106.36 (9.88)	Cadence (step·min^−1^)	100.06 (13.27)	95.11 (13.42)	91.97 (13.90)
Stride length (m)	1.29 (0.20)	1.27 (0.21)	1.25 (0.21)	Stride length (m)	1.11 (0.19)	1.06 (0.19)	1.02 (0.19)
Stride width (m)	0.14 (0.04)	0.15 (0.04)	0.15 (0.04)	Stride width (m)	0.14 (0.05)	0.15 (0.04)	0.15 (0.04)
Double support time (% GC)	31 (4.18)	32 (4.21)	32 (4.62)	Double support time (% GC)	34 (6.16)	36 (6.55)	37 (6.68)

EDSS, Expanded Disability Status Scale; GC, Gait Cycle

**Table 5 sensors-23-03407-t005:** Spatio-temporal variables during each interval of the 6MWT for PwMS with PP, RR, and SP phenotypes.

	PwMS with PP Phenotype (n = 14)			PwMS with RR Phenotype (n = 11)			PwMS with SP Phenotype (n = 20)		
				6MWT Intervals					
ST Variables	Initial	Middle	End	Initial	Middle	End	Initial	Middle	End
Velocity (m·s^−1^)	0.98 (0.31)	0.93 (0.32)	0.91 (0.31)	1.15 (0.22)	1.09 (0.25)	1.08 (0.20)	1.07 (0.26)	1.00 (0.25)	0.96 (0.27)
Cadence (step·min^−1^)	101.98 (15.16)	97.66 (15.49)	96.61 (15.46)	109.63 (7.00)	105.78 (6.88)	104.30 (7.02)	105.83 (14.12)	101.55 (13.96)	98.06 (14.62)
Stride length (m)	1.14 (0.27)	1.12 (0.27)	1.11 (0.28)	1.25 (0.20)	1.22 (0.23)	1.21 (0.20)	1.21 (0.17)	1.17 (0.18)	1.14 (0.19)
Stride width (m)	0.14 (0.03)	0.15 (0.04)	0.15 (0.04)	0.12 (0.03)	0.13 (0.03)	0.14 (0.03)	0.14 (0.06)	0.15 (0.05)	0.15 (0.05)
Double support time (% GC)	35 (7.31)	36 (7.91)	37 (7.85)	32 (4.27)	33 (4.40)	34 (4.44)	31 (4.07)	32 (4.25)	32 (4.72)

GC, Gait Cycle; PP, Primary Progressive; PwMS, People with Multiple Sclerosis; RR, Relapsing–remitting; SP, Secondary Progressive.

## Data Availability

We certify that this is an original manuscript with no plagiarism or illegal data fabrication, that has not been published or submitted to another journal and that no party having a direct interest in the results of the research has or will confer a benefit on us or on any organization with which we are associated.

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
