# Peer review of "Locomotor Strategy to Perform 6-Minute Walk Test in People with Multiple Sclerosis: A Prospective Observational Study"

_sensors, 2023, doi:10.3390/s23073407_

Round 1

Reviewer 1 Report

Dear authors, it was a pleasure to review this article. Despite its clinical relevance, the paper was well-written and well-structured. However, some minor changes can improve the paper.

Title

Although descriptevely interesting, the title could be improved by adding the study design.

Abstract

There is a typo error in line 28.

The abstract lacks information about methods, especially how ST measures were obtained. I strongly recommend adding this information.

The conclusion must also be improved with more specific results and potential contributions of this study for research and clinical practice.

Introduction

The introduction could benefit from being divided into paragraphs instead of being only a large one. It could be interesting for the readers to introduce some of the majors daily living activities impacts reported in literature. The first paragraph could end after this information and, then, the authors should start mentioning field tests.

I also recommend setting how field tests such as 6MWT were developed, as well as including the validation for PwMS.

The next paragraph should start in line 59, when the authors cited their own previous research.

Despite these previously mentioned aspects, I suggest reinforcing the need of this study and its potential contributions for research, clinical practice and patients functionality.

Methods

I have no considerations regarding methods section, except for including a sample size calculation. I only recommend a detailed revision for checking citations and possible minor spell errors.

Results

Table 2 should be described in detailed when reporting results. Were the comparisons within groups and intergroups performed for these variables?

The last paragraph can confuse the readers since the subgroup analysis was not mentioned. The authors should considering improve this information despite of including the supplementary data.

Discussion

In the first paragraph, I suggest resume the main purpose of this study followed by the

main findings - focusing in intragroup aspects - and their practical implications.

For a better understanding for the readers, the authors need to properly discuss the expected pattern and, then, explain the differences found. First, I suggest focusing only in intergroups differences. Then, the authors should emphasized the intragroups aspects, introducing their functionality implications. I also suggest a more in depth consideration regarding test's intensity (How cardiorespiratory fitness can help to understand the main findings?). It is a need better describe this aspect in results section as well.

I also recommend a detailed revision regarding English writing and an excessive and repetitive use of terms such as "Here (...)" and "Moreover (...)".

Although well-structured, the manuscript could benefit from adding limitations and strengths, as well as practical implications

and perspectives, as topics.

Conclusion

The conclusion must address the main findings from the study instead of being vague. I strongly recommend revising the conclusion.

Author Response

Locomotor strategy to perform 6-minute Walk Test in people with multiple sclerosis: a prospective observational study

The subject is adequate for the Special Issue of Sensors: Biomechanical Analysis of Motion and Postural Control: Sensors Methods and Data Analytics II.

Authors: we thank the reviewer and the editorial team for your comments on our work to improve it.

Please find below the replies labeled in blue color and the modifications have been highlighted in yellow in the manuscript.

Nevertheless, some aspects need further work, along with those asked by the reviewers.

Several acronyms are undefined when they first appear, for example, 6MWT in L2, L21 and EDSS in L92.

Authors: Thank you for your comment. We have revised all the manuscript. Each acronym was defined at its first utilization .

Yours previous article describes the walking pathway in the laboratory, but nevertheless you should provide a brief description of the procedures in “2.1. Study design”

Authors: Thank you for your comment, we have made precisions and completed the description of the procedures in the section « 3.2 walking evaluation »:

« Participants performed two 6MWTs (test and retest) one week apart in our laboratory; both tests were performed in accordance with instructions of the American Thoracic Society in the both groups [21], with the exception that the participants performed the test on a 24-m circuit rather than the recommended 32-m circuit. During the 6MWT, foot contacts were recorded by a 6.10 m by 0.61 m GAITRite electronic walkway (CIR Systems, Franklin, NJ, USA). At the same time, the software PKMAS (ProtoKinetics, Havertown, PA, USA) was used to process and export the following ST variables for each walkway passage: velocity (m/s), cadence (steps/min), stride length (m), stride width (m), and double support time (% of gait cycle). These measurements were recorded during 3 distinct intervals of 1 minute of the 6MWT [29]. Walking assessments in these specific intervals respect the number of steps required to study the aforementioned ST variables [29]. The studied intervals were: “initial” (from the start (0 seconds) to the end of the first minute (60 seconds); “middle” (from 2.5 min (150 sec-onds) to 3.5 min (210 seconds)); and “end” (from 5 min (300 seconds) to the end of the 6th minute (360 seconds)). These 1-minute intervals were chosen to offer a more discriminant insight in the analysis of strategies adopted during 6MWT while keeping appropriate levels of applicability and comprehensiveness for clinical practice [24,25]. To evaluate effort, a 15-grade Rating Perception Exertion Borg scale was performed during 6MWT at the end of each interval [30] and the total distance traveled at the end of 6MWT was measured. »

Since there are no limits on the number of tables or figures, please include in the main text all the information on the supplementary files.

Authors: Thank you for your comment and precisions about the absence of limit number of tables/figures. We have included figures 2 and 3 and tables 4 and 5 in the manuscript.

Reviewer one

Dear authors, it was a pleasure to review this article. Despite its clinical relevance, the paper was well-written and well-structured. However, some minor changes can improve the paper.

Title

Although descriptevely interesting, the title could be improved by adding the study design.

Authors: Thank you for your comment, we changed the title for « Locomotor strategy to perform 6-minute Walk Test in people with multiple sclerosis: a prospective observational study »

Abstract

There is a typo error in line 28.

The abstract lacks information about methods, especially how ST measures were obtained. I strongly recommend adding this information.

The conclusion must also be improved with more specific results and potential contributions of this study for research and clinical practice.

Authors: Thank you for your comment, we corrected the typo error « pattern » ( line 30, page 1) and we have added information in the abstract as recommended :

« Participants performed a 6MWT with measures of 5 ST variables during three 1-min intervals (initial: 0′–1′, middle: 2′30″–3′30″, end: 5′–6′) of the 6MWT, using the GAITRite system .» (lines 23-25, page 1)

« Beyond walking distance measure, other ST parameters measures during specific keys intervals of 6MWT could easily be used in clinical routine practice with the emergence of wearable PST recordable systems. These parameters  have allowed to better understand PwMS’ compensations to deal with prolonged walking conditions. Further studies are needed to detect those patterns in the early evolution of disease, identify clinical determinants involved in PwMS’ walking pattern, and to investigate whether interventions can have a positive impact on this pattern. » (lines 32-38, page 1)

Introduction

The introduction could benefit from being divided into paragraphs instead of being only a large one. It could be interesting for the readers to introduce some of the majors daily living activities impacts reported in literature. The first paragraph could end after this information and, then, the authors should start mentioning field tests.

Authors: Thank you for your comment, we have improved and detailed this part as recommended.

« The symptoms of MS are variable, but most of them are sensory, cognitive, and motor impairments. It has a negative impact on the activity and participation of People with MS (PwMS) in the meaning of International Classification of Functioning disability and health, with disabilities in daily-life activities e.g., to walk, to grip, to speak, to have a good bladder and bower control or a satisfying sexual life, or even visual, thinking, memory and mood disabilities [3–5]. Among all of them, for the majority of Persons with MS (PwMS), walking disability is the most limiting factor for daily-life activities [3][6]. This is also the reason why walking disability is a major criterion to assess the MS progression in the Expanded Disability Severity Scale (EDSS) [7]. The EDSS has been considered as the gold standard for diagnosing the clinical and functional severity of MS [7]. However, recent studies have highlighted its limitations and suggested that this scale should be supplemented by other objective standardized measurements [8,9].” (lines 45-57).

I also recommend setting how field tests such as 6MWT were developed, as well as including the validation for PwMS.

Authors: Thank you for your comment, we have added these information with references : Thank you for your comment, we have added these information with references: « Initially, the 6MWT has been commonly used for a long time in clinical assessment of exercise capacity in patients with cardiopulmonary diseases [19,20] and was validated in MS, and recommended in a clinical guideline to assess MS since 2018 [11,12]. » (lines 75-78).

The next paragraph should start in line 59, when the authors cited their own previous research.
Authors: Thank you for your comment, we have made as you can see several paragraphs as recommended.

Despite these previously mentioned aspects, I suggest reinforcing the need of this study and its potential contributions for research, clinical practice and patients functionality.

Authors: Thank you for your comment, we have added the sentence below.

« The assessment of other ST parameters measures during specific keys intervals of 6MWT, which could be easy-to-use in clinical routine practice; will allow to more understand impossibilities of loss compensation during prolonged walking conditions and will permit to determine walking dysfunction earlier than classic analyze in a degenerative disease. » (lines 95-99).

Methods

I have no considerations regarding methods section, except for including a sample size calculation. I only recommend a detailed revision for checking citations and possible minor spell errors.

Authors: thank you for your comment. We added these elements to the methods section concerning the sample size calculation: “Due to the aim of our previous study, the power was verified for the available sample of 45 PwMS concerning the evaluation criterion of the reliability of the ST pattern during the 6MWT. Assuming a minimal ICC of 0.5 against a desired of 0.8 based on α = 0.05, n (number of observations) = 2, power was 99.3% [32].” (lines 167-170).

We included to the discussion section limitation about the sample size: “Finally, the sample size was a limitation. Indeed, it was based on the previous study aim i.e., reliability study. Moreover, our sample size was precluded any investi-gation of the relationship between ST walking pattern and MS characteristics. For fu-ture studies, it will be important to consider the variation for a given criterion (e.g., mean of standard deviations for speed intervals) and a minimum expected change be-tween intervals (e.g., at least the same change observed in the healthy group).” (lines 405-410)

Results

Table 2 should be described in detailed when reporting results. Were the comparisons within groups and intergroups performed for these variables?

Authors: thank you for your questions. We described more results concerning the table 2 as recommended:

« In PwMS, a monotonous decrease pattern was observed for velocity, cadence and stride length during the 6MWT . Stride width and double support in the PwMS group showed a constant increase. In healthy group, velocity, cadence and stride length decreased between the initial and middle intervals and increased between the middle and final intervals. No major variation was shown for stride width between the intervals and an increase from initial to middle interval (Table 2). » (lines 193-196)

The comparisons were performed for the changes and standard error of measurement between middle/initial and final/initial within groups (e.g., PwMS and healthy groups) and between groups. We clarified it in the results: «Table 3 summarized the changes and standard error of measurement between middle/initial and final/initial in each group (e.g., PwMS and healthy groups) and between groups. » 

Concerning the variables presented in the table 2, differences between groups and between intervals were examined by an analysis of variance (ANOVA) with 2 factors, namely group (i.e., PwMS and healthy subjects) and interval (i.e., the 3 intervals of 6MWT) :

- In PwMS, a significant monotonous decrease was observed for velocity, cadence and stride length during the 6MWT (p-values ranging from 0.20x104 to 0.002), albeit without a significant decrease from the middle to the end interval (p-values ranging from 0.09 to 0.55). Stride width and double support in the PwMS group showed a constant significant increase (p-values ranging from <0.0001 to 0.04).

- In healthy group, velocity, cadence and stride length decreased between the initial and middle intervals and increased between the middle and final intervals. The variations were statistically significant for velocity and cadence from the initial to middle interval (p-values ranging from 0.0086 to 0.0001) and were significant only for velocity from the middle to end interval (p-values ranging from 0.0001 to 0.02). The observed variations were not significant for stride length. Moreover, no significant variation was shown for stride width and double support time (p-values ranging from 0.08 to 1), except for a significant increase for the double support from the initial to the middle interval (p-value = 0.0086).

But the results were not included in the present paper because since velocity differed between groups (Table 2 and previous article) and that literature shows that velocity is strongly related to ST parameters, for statistical analysis, ST parameter values were expressed in changes (%) in relation of the values observed in the initial interval. However, if you think it would be interesting to add these analyses, we can add a table but we thought that the statistical analyses of the changes due to the above explanations were enough so as not to confuse the readers.

We clarified in the statistical analysis section why we performed comparison with changes (%) variables and not with variables presented in the table 2:

« Since velocity differed between groups and that literature shows that velocity is strongly related to ST parameters, ST parameter values were expressed in changes (%) in relation of the values observed in the initial interval ». (lines 172-174).

The last paragraph can confuse the readers since the subgroup analysis was not mentioned. The authors should considering improve this information despite of including the supplementary data.

Authors: Thank you for your comment. We added information and descriptions to be more understanding:

« In sub-groups of PwMS, we could observe some trends of the descriptive course of the ST walking pattern in the intervals of the 6MWT according to the EDSS status (moderate (EDSS of 4) and severe (EDSS between 4.5 and 6.5)) (Table 4 and Figure 2) [12] and according to MS phenotype (Table 5 and Figure 3). We observed a more pronounced monotonous decrease in velocity between the middle and end intervals in patients with severe EDSS, compared to moderate EDSS. Velocity, cadence and stride length were lower in each interval in those with severe EDSS compared to the moderate EDSS group, while stride width and double support time were higher in the severe EDSS group than in the moderate EDSS group. Velocity, cadence and stride length in the SP group showed a more marked decrease than the other 2 groups, which showed a monotonous decrease. Velocity, cadence, and stride length were lower in each interval in the PP group than in the other 2 groups, while stride width and double support time were higher in the PP group than in the other 2 groups. Velocity, cadence, and stride length were lower in each interval in the SP group than in the RR group, while stride width and double support time were higher in the SP group than in the RR group. » (lines 234-248).

The supplementary data were including in the manuscript (e.g., Figures 2 and 3, Table 4 and 5).

Discussion

In the first paragraph, I suggest resume the main purpose of this study followed by the

main findings - focusing in intragroup aspects - and their practical implications.

For a better understanding for the readers, the authors need to properly discuss the expected pattern and, then, explain the differences found. First, I suggest focusing only in intergroups differences. Then, the authors should emphasized the intragroups aspects, introducing their functionality implications. I also suggest a more in depth consideration regarding test's intensity (How cardiorespiratory fitness can help to understand the main findings?). It is a need better describe this aspect in results section as well.

I also recommend a detailed revision regarding English writing and an excessive and repetitive use of terms such as "Here (...)" and "Moreover (...)".

Although well-structured, the manuscript could benefit from adding limitations and strengths, as well as practical implications and perspectives, as topics.

Conclusion

The conclusion must address the main findings from the study instead of being vague. I strongly recommend revising the conclusion.

Authors: thank you for your comment. We clarified some results and perspectives of our study in the conclusion as recommended: « To conclude, analysis of ST pattern throughout specific intervals on 6MWT revealed that PwMS have an altered walking pattern (i.e., a constant decline pattern) of velocity, cadence and stride length compared to healthy persons who displayed a rebound pattern but without significant changes during the 6MWT. These results reflecting an altered walking management in PwMS despite the fact that the velocity or the 6MWT distance travelled in PwMS were within the normal limits values of healthy people. In addition to the velocity pattern in PwMS, stride width and double support time increased during the 6MWT. This could be correlated with the decrease of walking velocity, cadence and stride length, and reflect a loss of walking efficacy. These results underlined the importance of integrating variables that reflect walking stability, to obtain a better understanding of the overall ST pattern during the 6MWT in PwMS. With the better accessibility to the wearable ST analysis systems, this type of analysis will be easy-to-use in clinical routine practice. The link between retrieving pattern and the degree of impairment allows us to put this analysis in an early detection of gait impairment. Further studies are needed to identify clinical determinants involved in this altered pattern, and to investigate whether interventions can have a positive impact on this pattern (i.e., by stimulating PwMS to move from a constant decline pattern to a rebound or constant ST pattern), and not just focus on the improvement of the velocity or 6MWT distance travelled to improve walking capacities of PwMS ». (lines 412-430)

Reviewer 2 Report

Dear Authors,

I would like to thank you for conducting this nice study. I have enjoyed reading your manuscript, however; I have some major concerns as follows:

Introduction:

Line 37: The authors need to re-write this sentence "Walking disorder in Persons with MS (PwMS) is the most limiting factor for activity". Which activity is limited by walking disorder? Is the walking disorder the primary cause leading to activity limitations or other factors contributing to walking impairment?

I would replace "Persons" with Individuals or People.

Line 49-53: The authors used alternatively walking, activity and performance. These terms are different and they are not interchangeable. The authors need to be clear about which variable to be measured. The 6MWT is not a laboratory test, and it is not intended to evaluate the walking or gait or reflect everyday performance.

Line 59: The authors reported "we published a first study based on evaluate test-retest reliability of ST parameters in some key intervals of the 6MWT in 60 PwMS, which were reliable", where is the reference? what was the results of the reliability?

Methods

How did you determine the capacity to walk for a period of at least 6 minutes? Did individuals with MS suffer from any cardiorespiratory disease or other neurological disorders?

What an EDSS? why did you choose a score of 4.0–6.5?

Line 96: How did you recruit healthy persons? How did you determine if they were healthy considering their age?

Line111-113: Did the authors conduct the test inside the lab? What was the walkway length? Was the whole walkway covered  with GAI-TRite electronic walkway system (CIR Systems, Franklin, NJ, USA). 

Line 116-120: The authors divided the intervals into the following categories: “initial” (from the start (0 seconds) to the end of the first minute (60 seconds); “middle” (from 2.5 min (150 seconds) to 3.5 min (210 seconds)); and “end” (from 5 min (300 seconds) to the end of the 6th minute (360 seconds)). Based on what did the authors choose these stratifications?

How did you determine the sample size?

Results:

The authors found the speed in the middle phase (between 2.5 to 3.5 m) was lower than the first phase, which is opposite the literature. Usually, the speed during the 6MWT increases after the second minute as the momentum of the participant increases. How would the authors explain this reduction in the speed?

The results of the Borg scale showed that the test was suboptimal and the participants in the healthy group did not reach the submaximal level, as it should be for the 6MWT. Subsequently, the distance achieved is questionable.

In table 3, the authors presented the changes as % and SEM. The SEM was large for most of the variables, indicating huge variability between the participants. This does not match with the numbers in table 2. Could the authors explain these variabilities? I would prefer if the authors report the 95%CI instead of SEM.

Line 178-186: The authors conducted a within-group analysis, which apparently showed no significant differences between the ST variables. First, the authors need to define the severity of the disease in the methods based on EDSS. Second, I would like to see the differences between the healthy group and the disease phenotype.

Discussion:

This part could be rewritten and improved after addressing the previous comments.

The velocity and stride length of individuals with MS are still within the normal limits of healthy individuals. The distance covered in the 6MWT is still considered above the minimum acceptable limit for individuals with MS.

Sincerely,

Author Response

Locomotor strategy to perform 6-minute Walk Test in people with multiple sclerosis: a prospective observational study

The subject is adequate for the Special Issue of Sensors: Biomechanical Analysis of Motion and Postural Control: Sensors Methods and Data Analytics II.

Authors: we thank the reviewer and the editorial team for your comments on our work to improve it.

Please find below the replies labeled in blue color and the modifications have been highlighted in yellow in the manuscript.

Reviewer two

Dear Authors,

I would like to thank you for conducting this nice study. I have enjoyed reading your manuscript, however; I have some major concerns as follows:

Introduction:

Line 37: The authors need to re-write this sentence "Walking disorder in Persons with MS (PwMS) is the most limiting factor for activity". Which activity is limited by walking disorder? Is the walking disorder the primary cause leading to activity limitations or other factors contributing to walking impairment?

Authors: Thank you for your comment, we have added some data for better understanding for the readers: “The symptoms of MS are variable, but most of them are sensory, cognitive, and motor impairments. It has a negative impact on the activity and participation of People with MS (PwMS) in the meaning of International Classification of Functioning disability and health, with disabilities in daily-life activities e.g., to walk, to grip, to speak, to have a good bladder and bower control or a satisfying sexual life, or even visual, think-ing, memory and mood disabilities [3–5]. Among all of them, for the majority of Per-sons with MS (PwMS), walking disability is the most limiting factor for daily-life activ-ities [3,6]. This is also the reason why walking disability is a major criterion to assess the MS progression in the Expanded Disability Severity Scale (EDSS) [7]. The EDSS has been considered as the gold standard for diagnosing the clinical and functional severi-ty of MS [7]. However, recent studies have highlighted its limitations and suggested that this scale should be supplemented by other objective standardized measurements [8,9].” (lines 45-57).

I would replace "Persons" with Individuals or People.

Authors: Thank you for your comment, we have made the modification: « It has a negative impact on the participation of People with MS (PwMS) in … » (lines 46-47).

Line 49-53: The authors used alternatively walking, activity and performance. These terms are different and they are not interchangeable. The authors need to be clear about which variable to be measured. The 6MWT is not a laboratory test, and it is not intended to evaluate the walking or gait or reflect everyday performance.

Authors: Thank you for your comment, we have made modifications to be more precise:

« Although there is no clear consensus on which test should be used to assess walking in PwMS, the 6-minute walk test (6MWT) was recently recommended to evaluate gait in PwMS, because it highlight motor fatigue resulting from extended task execution, thus, effectively assessing the physical efforts and level of autonomy of PwMS. For these reasons, the prolonged tests as the 6MWT are a better indicator of the ability of PwMS to perform the activities of daily living [19,20]. » (lines 70-75).

Line 59: The authors reported "we published a first study based on evaluate test-retest reliability of ST parameters in some key intervals of the 6MWT, which were reliable", where is the reference? what was the results of the reliability?

Authors: Thank you for your question, we specified the results concerning our previous study published in Sensors: « In this context, we published a first study based on evaluate test-retest reliability of ST parameters in some key intervals of the 6MWT in PwMS, which were reliable (e.g., intraclass correlation coefficient (ICC) range for PwMS: 0.858–0.919) [25]. » (lines 85-87).

Methods

How did you determine the capacity to walk for a period of at least 6 minutes?

Authors: Thank you for your question, it was according to the data from the medical examination and by performing an EDSS.

Did individuals with MS suffer from any cardiorespiratory disease or other neurological disorders?

Authors: Thank you for your question, indeed the PwMS not suffer from any significant chronic cardiorespiratory disease or other neurological disorders. We specified these elements in the participants’ inclusion exclusion criteria in the methods section:

« All participants did not suffer from any significant chronic cardiorespiratory disease. »

« (iii) the capacity to walk for a period of at least 6 minutes (according to the data from the medical examination i.e., EDSS). »

« The PwMS did not suffer from other neurological disorders. » (lines119-131)

What an EDSS? why did you choose a score of 4.0–6.5?

Authors:  Thank you for your questions. EDSS is for Expanded Disability Severity Scale (please see Kurtzke, J.F. Rating Neurologic Impairment in Multiple Sclerosis: An Expanded Disability Status Scale (EDSS). Neurology 1983, 33, 1444–1452, doi:10.1212/wnl.33.11.1444.) and is the main scale used to assess the level of MS severity. We clarified it in the introduction when it first appeared.

We included PwMS in this range because it is between the values of 4.0 and 6.5 that gait worsening is rated. We clarified it in the methods section as recommended.

« (ii) We included PwMS in this range because it is between the values of 4.0 and 6.5 that gait worsening is rated [7]. According to the literature, PwMS with an EDSS of 4 had a moderate MS. PwMS with an EDSS between 4.5 and 6.5 had a severe MS [12] ; (…) » (lines 123-126).

Line 96: How did you recruit healthy persons? How did you determine if they were healthy considering their age?

Authors:  Thank you for your questions. A group of healthy volunteers (healthy group) was recruited from the general community to make a comparable group with PwMS in terms of distribution of sex, age, body mass, body height, and body mass index (BMI); they presented no neuro-orthopedic problems or other antecedents that could compromise their walking capacities. We clarified it in the section methods:

« A group of healthy volunteers (healthy group) was recruited from the general community to make a comparable group with PwMS in terms of distribution of sex, age, body mass, body height, and body mass index (BMI); they presented no neuro-orthopedic problems or other antecedents that could compromise their walking capacity. » (lines 132-135)

Line111-113: Did the authors conduct the test inside the lab? What was the walkway length? Was the whole walkway covered  with GAITRite electronic walkway system (CIR Systems, Franklin, NJ, USA). 

Authors: Thank you for your questions. Yes the 6MWT was conduct inside our laboratory in a 24-m circuit. We clarified it as recommended: « Participants performed two 6MWTs (test and retest) one week apart in our laboratory; both tests were performed in accordance with instructions of the American Thoracic Society in the both groups [21], with the exception that the participants per-formed the test on a 24-m circuit rather than the recommended 32-m circuit due to local architectural constraints. » (lines 147-151)

Only 25% of this length was explored by GAITRite electronic walkway system. We clarified it in the discussion section: « only 25% of this length was explored by the GAITRite electronic walkway system ». (lines 389-390). 

Line 116-120: The authors divided the intervals into the following categories: “initial” (from the start (0 seconds) to the end of the first minute (60 seconds); “middle” (from 2.5 min (150 seconds) to 3.5 min (210 seconds)); and “end” (from 5 min (300 seconds) to the end of the 6th minute (360 seconds)). Based on what did the authors choose these stratifications?

Authors: thank you for your question. These intervals were chosen to better explore and distinguish the different strategical periods of adaptation and tolerance involved in performing a 6MWT, as we have reported previously in an other disease [20].

We clarified it in the methods section: “These 1-minute intervals were chosen to offer a more discriminant insight in the anal-ysis of strategies adopted during 6MWT while keeping appropriate levels of applicability and comprehensiveness for clinical practice [24,25].” (lines 161-164)  

How did you determine the sample size?

Authors: thank you for your question. We added these elements to the methods section concerning the sample size calculation: “Due to the aim of our previous study concerning the reliability of the ST pattern of PwMS during the 6MWT, the study power was verified for the available sample of 45 PwMS. Assuming a minimal ICC of 0.5 against a desired of 0.8 based on α = 0.05, n (number of observations) = 2, power was 99.3% [32]. 

We included to the discussion section limitation about the sample size: “Finally, the sample size was a limitation. Indeed, it was based on the previous study aim i.e., reliability study. Moreover, our sample size was precluded any investigation of the relationship between ST walking pattern and MS characteristics. For future studies, it will be important to consider the variation for a given criterion (e.g., mean of standard deviations for speed intervals) and a minimum expected change be-tween intervals (e.g., at least the same change observed in the healthy group).” (lines 405-410)

Results:

The authors found the speed in the middle phase (between 2.5 to 3.5 m) was lower than the first phase, which is opposite the literature. Usually, the speed during the 6MWT increases after the second minute as the momentum of the participant increases. How would the authors explain this reduction in the speed?

Authors: Thank you for your question. Several articles in the literature have shown a decrease in walking velocity in healthy people during an instrumented 6MWT between the beginning of the 6MWT and the middle phase of this test.

Please see these different studies for example with healthy controls and PwMS (which only assessed the velocity during the 6MWT but not other ST parameters):

Goldman MD, Marrie RA, Cohen JA. Evaluation of the six-minute walk in multiple sclerosis subjects and healthy controls. Mult Scler. 2008 Apr;14(3):383-90. doi: 10.1177/1352458507082607. Epub 2007 Oct 17. PMID: 17942508.

Chen S, Sierra S, Shin Y, Goldman MD. Gait Speed Trajectory During the Six-Minute Walk Test in Multiple Sclerosis: A Measure of Walking Endurance. Front Neurol. 2021 Jul 26;12:698599. doi: 10.3389/fneur.2021.698599. PMID: 34381416; PMCID: PMC8352578.

Please see these different studies for example with healthy controls and people with other diseases as MS:

  1. Hadouiri, D. Feuvrier, J. Pauchot, P. Decavel, Y. Sagawa, Donor site morbidity after

vascularized fibula free flap: gait analysis during prolonged walk conditions, Int. J. Oral

Maxillofac. Surg. 47 (2018) 309–315. https://doi.org/10.1016/j.ijom.2017.10.006.

  1. Beausoleil, L. Miramand, K. Turcot, Evolution of gait parameters in individuals with a

lower-limb amputation during a six-minute walk test, Gait Posture. 72 (2019) 40–45.

https://doi.org/10.1016/j.gaitpost.2019.05.022.

This reduction of walking velocity seems to be in agreement with the bioenergetic evolution pattern such as the evolution of VO2max or heart rate during a 6MWT for healthy people (Mänttäri A, Suni J, Sievänen H, Husu P, Vähä-Ypyä H, Valkeinen H, Tokola K, Vasankari T. Six-minute walk test: a tool for predicting maximal aerobic power (VO max) in healthy adults. Clin Physiol Funct Imaging. 2018 May 31. doi: 10.1111/cpf.12525. Epub ahead of print. PMID: 29851229.)

The results of the Borg scale showed that the test was suboptimal and the participants in the healthy group did not reach the submaximal level, as it should be for the 6MWT. Subsequently, the distance achieved is questionable.

Authors: Thank you for your comment. The Borg scale used in our laboratory was a 15-grade scale (e.g., Rating of Perceived Exertion Borg scale). The instruction of the 6 minutes is to walk as fast as possible for 6 minutes. The Borg scale reflects the exertional intensity of resistance exercise. Thus, the subjects were able to give the maximum (sub-maximum) speed without feeling the maximum exertion. The interest here is to show that in subjects with MS, there is a painfulness of walking that should not be felt as such, in the absence of resistance to effort. We clarified it after your comment in the methods section: « a 15-grade Rating Perception Exertion Borg scale was performed during 6MWT at the end of each interval [31]». (lines 164-165).

We also modified the discussion:

PwMS presented in our study (i) a “constant decline pattern” with significant changes between initial and final intervals of 6MWT, which were superior to SEM and (ii) higher scores in Borg scale so a perceived exertion more negative than healthy people. The Borg scale was designed to assess the strenuousness of a physical activity in resistance, which is not the case for walking. However, there is an increased exertion in MS patients compared to healthy subjects during a submaximal task such as the 6-minute test. However, we had shown that the reliability of the Borg scale was not good in MS subjects in our previous article [25].(lines 324-331).

The literature reported the 6MWT as a submaximal test according to the American Thoracic Society and the distance achieved in our study by healthy people were assimilated to those presented in the literature.

In table 3, the authors presented the changes as % and SEM. The SEM was large for most of the variables, indicating huge variability between the participants. This does not match with the numbers in table 2. Could the authors explain these variabilities? I would prefer if the authors report the 95%CI instead of SEM.

Authors: thank you for your question but we are not sure if we understood it correctly. The SEMs were also expressed as a percentage so they are not that high (please see the table 3). The CIs are already put in the previous reliability article. But we can reintroduce them in the table 3.

If the answer was not what you expected, could you clarify your question? Thank you very much.

Line 178-186: The authors conducted a within-group analysis, which apparently showed no significant differences between the ST variables. First, the authors need to define the severity of the disease in the methods based on EDSS. Second, I would like to see the differences between the healthy group and the disease phenotype.

Authors: Thank you for your comment. We clarified that significant deteriorations were observed for all ST variables except stride width between initial and final intervals in the PwMS with changes superior to SEM and for healthy people, changes were inferior to SEM: « In PwMS, statistical significant deteriorations were observed for all ST variables except stride width between initial and final intervals. In relation to initial interval and for all ST variables, changes varied between 2% and 9%. Changes were superior the SEM only in the PwMS group between initial and final interval and varied between 2.6 and 8.4 for this group and intervals (Table 3) (…) but in healthy people (…) changes varied between 0% and 3% and were inferior to SEM in all intervals comparisons (SEM values ranged from 2.5 to 7.1) (Table 3). »  (lines 210-224).

We clarified in the manuscript the severity of the disease in the methods section as recommended:

« We included PwMS in this range because it is between the values of 4.0 and 6.5 that gait worsening is rated [7]. According to the literature, PwMS with an EDSS of 4 had a moderate MS. PwMS with an EDSS between 4.5 and 6.5 had a severe MS [12] » (lines 124-126).

We have included in the manuscript differences between healthy people and PwMS in the figure 1. Concerning trends of differences between groups of MS phenotype (Figure 3), we have included after your recommandations a figure but without healthy people because velocity differences were very important in terms of scale, and if we made in a same figure thevalocity pattern during the 6MWT of healthy people and the differents groups of MS phenotype, differences between MS phenotype groups could be not seen.

If the answer was not what you expected, could you clarify your question? Thank you very much.

Discussion:

This part could be rewritten and improved after addressing the previous comments.

The velocity and stride length of individuals with MS are still within the normal limits of healthy individuals. The distance covered in the 6MWT is still considered above the minimum acceptable limit for individuals with MS.

Authors: Thank you for your comment. After taking into account your suggestions and those of reviewer 1, we have restructured the discussion to make it easier to understand the different elements.

Round 2

Reviewer 2 Report

Dear Authors,

The manuscript is difficult to read and follow. The authors need to make extensive editing for their manuscript.

The authors have not sufficiently addressed the response to the concerns.

Best regards,

Author Response

Dear Reviewer,
please find the new version of the manuscript with modifications
according to the latest recommendations and questions from the editor-in-chief.
We hope you will find the manuscript clearer.

Best regards